# The commercial promotion of electronic cigarettes on social media and its influence on positive perceptions of vaping and vaping behaviours in Anglophone countries: A scoping review

L. Chacon, G. Mitchell¤, S. Golder*

Department of Health Sciences, University of York, Heslington, United Kingdom

¤ Current address: Institute of Social Marketing and Health, University of Stirling, Stirling, Scotland, United Kingdom
* su.golder@york.ac.uk

**Data Availability Statement:** The included studies are all published. The supplementary material includes full details of our search strategies. Our

## Abstract

There is ongoing scientific and policy debate about the role e-cigarettes play in tobacco control, with concerns centring around unknown long-term effects, and the potential industry co-option of harm reduction efforts, including marketing to youths. There is substantial evidence of the influence of conventional cigarette promotion on smoking behaviours in Anglophone countries, and the popularity of social networking sites, as well as the lack of marketing regulations on the commercial promotion of electronic cigarettes online, suggest an urgent need to explore this topic further. This scoping review aims to map the existing evidence related to the influence of e-cigarette commercial promotion on social media on positive perceptions of vaping and vaping behaviours in core Anglophone countries. Searches were conducted in CENTRAL, Cochrane Database of Systematic Reviews (CDSR), Embase, Epistemonikos, MEDLINE, PsycINFO and Science Citation Index, on the 21st of July 2022. From 1,385 studies, 11 articles were included in the final review, using diverse study designs, including focus groups, content analysis, cross-sectional studies, and experiments. The studies were primarily based in the U.S. and evidenced the association between the commercial promotion of e-cigarettes on social media with positive perceptions of vaping and vaping behaviours, particularly among young people, addressing diverse themes including celebrities' sponsorship, e-liquid appeal (including flavours and nicotine levels), users' engagement with ads, and other marketing strategies. Further, social networking sites commercially promoting e-cigarettes might increase positive attitudes towards vaping and vaping behaviours, particularly among youths. Future research should be conducted in broader settings, incorporate larger and diverse sample sizes, ensure research transparency, cover multiple social networking sites, emphasize ecological validity, and foment longitudinal studies.

tables include all extracted information. All relevant data are within the manuscript and its Supporting information files.

**Funding:** The authors received no specific funding for this work.

**Competing interests:** The authors have declared that no competing interests exist.

## Introduction

Differently from conventional cigarettes, electronic cigarettes (or e-cigarettes) heat a liquid substance to create an inhalable vapour, instead of burning tobacco [1]. This liquid, commonly referred to as e-liquid, contains various flavours, additives, and chemicals that can be harmful to human health [2]. E-cigarettes can either contain nicotine or be nicotine-free [2] and generally contain fewer harmful substances compared to conventional cigarettes [3]. Therefore, e-cigarettes are commonly considered a healthier alternative to traditional cigarette smoking [4].

The role of e-cigarettes in tobacco control is subject to significant policy debate. While some evidence suggests that e-cigarette contributes to smoking cessation [5], there is also concern that their use may lead to increased smoking behaviours among young people [6–8], potentially serving as a gateway to smoking [9]. Hence, the paradox surrounding e-cigarette usage is related to the harm reduction benefit it promotes for former smokers against the health hazards they present, including for non-smokers [10, 11].

The indiscriminate use of electronic cigarettes poses many health risks to humans, especially to the respiratory system [4], as observed during the E-cigarette or Vaping Use-associated Lung Injury (EVALI) epidemic outbreak in 2019 in the U.S. [12–14]. In core Anglophone nations such as the U.S., the use of e-cigarettes (also known as 'vaping') is a widespread practice among young individuals [15–17], who are often the target of the vape industry's appealing marketing strategies [18].

As observed previously in the tobacco industry marketing strategies, cigarette promotion in traditional communication media channels such as television, radio, and movies played a significant role in smoking behaviours [19]. Similarly, exposure to electronic cigarettes on social networking sites (also known as social media) has been associated with vaping behaviours across diverse types of smoking statuses individuals [20–23]. Social media is widely recognized as internet-based channels that enable individuals and companies to interact with a broad spectrum of audiences and create and share online content [20, 24]. In the context of e-cigarettes, social media can be accessed by users to obtain information about the product. Through the widespread dissemination of information across these media platforms, including targeted company content to specific demographics, the likelihood of users adopting e-cigarettes increases [20].

The validated correlation of the promotion of conventional cigarettes in traditional media on smoking behaviours [19] contributed to long-term restrictive marketing policies [25], resulting in a decrease in smokers' awareness of pro-smoking cues [26], and a reduction of conventional cigarette use [27]. Due to the rising concern about vaping [28], recent sessions at the World Health Organization Framework Convention on Tobacco Control (WHO FCTC) considered incorporating regulatory measures restricting e-cigarettes [28], including its manufacture, importation, distribution, presentation, sale and use [29]. Despite the WHO's efforts to regulate Electronic Nicotine Delivery Systems and Electronic non-nicotine Delivery Systems (ENDS/ENNDS) [29], current actions appear to be insufficient in effectively restricting their exposure on social networking sites [30]. Hence, enforcement of regulations on e-cigarette advertising, promotion and sponsorship might be lacking.

Regulatory frameworks for e-cigarette advertising on social media platforms may also vary between countries, as they fall under distinct jurisdiction policies [31]. In Australia, for example, e-cigarette marketing is completely prohibited [32], whereas in the U.S., the FDA has introduced various restrictions on e-cigarette marketing on social media [30]. Several self-regulatory restrictions by social media platforms that prohibit the promotion of tobacco products also fail to address novel forms of promotion, including sponsored content and controlling underaged access [30]. Industry influence over science and policy is an ongoing concern,

particularly, due to the tobacco industry's stake in, and promulgators of various e-cigarette brands [33, 34]. This influence is evidenced in studies with industry-related conflicts of interest frequently promoting vaping as a harmless activity [35].

More comprehensive policies are therefore necessary to restrict e-cigarette advertising on social media, and it is imperative to continue researching this phenomenon in the context of e-cigarette promotion in the 21st century. Despite the existing evidence regarding the impact of e-cigarette promotion on social media platforms on vaping behaviours [20–23], further rigorous research methods are necessary to establish the direct association of this exposure with vaping practices.

The primary objective of this scoping review was to synthesize the available evidence and identify key knowledge gaps, to support future investigations. Specifically, this review focused on exploring the influence of commercial e-cigarette promotion on social networking sites on the development of positive perceptions of vaping and various vaping behaviours (including experimentation, initiation, and escalation) across key Anglophone countries.

## Materials and methods

The scoping review is reported according to the Preferred Reporting Items for Systematic reviews and Meta-Analyses extension for Scoping Reviews (PRISMA-ScR) guidance [36] and guided by Arksey and O'Malley's framework underpinned by five methodological steps, developed with an iterative and reflexive approach [37]. In addition to the Arksey and Malley framework for scoping reviews, this study was also guided by recommendations provided by Levac and collaborators, to enhance a team-based approach to scoping review methodologies [38]. The methodology of the research was based on a previously published study protocol [39].

### Objective

This scoping review (1) explores the existing evidence related to the association between the commercial promotion of electronic cigarettes on social media and its influence on vaping initiation and positive perception of vaping in Anglophone countries, through a diverse range of study designs (2) maps the existing literature to determine whether systematic reviews related to the research topic are available or feasible to be developed, or whether primary research is first required (3) describes, summarizes, and disseminates in detail the key concepts and findings related to the research topic of interest, contributing to data accessibility (4) establishes what prospective research must cover, by identifying the main gaps in the literature related to the research topic, and contributing to a long-term, improved understanding of the theme.

### Identifying the research question

This review aimed to address the following research questions: 1) What evidence is there that the commercial promotion of electronic cigarettes on social media influences positive perceptions about vaping and vaping behaviours among individuals from Anglophone countries? and 2) What further research is needed to evidence the association between the commercial promotion of electronic cigarettes on social media and vaping initiation and positive perceptions about vaping among individuals from Anglophone countries?

### Identifying relevant studies

Seven databases were searched in the fields of health sciences, public health, social sciences, and psychology. The databases were selected due to their relevance to the research question. The databases were the Cochrane Database of Systematic Reviews (CDSR), Cochrane Central

**Table 1. PECO framework to structure the research question.**

| PECO | SCOPING REVIEW DEFINITION |
|---|---|
| *Population* | Individuals over 10 years old, from core Anglophone countries. |
| *Exposure* | Commercial promotion of electronic cigarettes on popular social media. |
| *Comparator* | Electronic cigarette brands, different flavours, celebrity-endorsed promotions, different social media formats and non-exposure. |
| *Outcome* | Positive perceptions of vaping and vaping behaviours (e.g., vaping experimentation, onset, and escalation) |

Register of Controlled Trials (CENTRAL), Epistemonikos, Medline, Embase, PsycINFO, and Science Citation Index (SCI). By following Cochrane guidelines [40], an additional search was conducted within previous reviews on equivalent topics and reference lists of the included studies were checked. Diverse combinations of text words and indexing terms and their respective synonymous, variations, and abbreviations were used in the search to identify relevant studies. The terms were chosen based on the PECO question formulation framework (Table 1) and from previous search strategies applied in topic-related systematic reviews [41–43]. Databases were searched until the 21st of July 2022. No language or date restrictions were applied.

## Selecting the studies

Given the current state of concern with e-cigarette use across the globe [2], it's worth noting that this scoping review specifically focuses on core Anglophone countries where English serves as the primary spoken language. This decision is rooted in practical considerations, including the language accessibility and data availability of studies in English. Moreover, this selection criterion is based on the substantial influence employed by Anglophone countries—such as the U.S.—over global regulatory discussions concerning e-cigarettes, increased funding for tobacco research, policy advocacy, and international cooperation.

Although it is not usual practice to contain commercial affiliations, user-generated content can carry undertones of commercial ties, which are not always declared [44]. However, in the absence of disclosed paid relationships, user-generated content may be interpreted as organic posts, and consequently, exempt from advertising restrictions [44]. While acknowledging the impact of excluding user-generated content, this research limits its scope to posts disclosing commercial ties.

Thematic and content analysis of social media significantly contributes to understanding how e-cigarettes are portrayed online [45]. However, merely including a descriptive analysis of e-cigarette commercial content on social media may fail to address its impact on users' favourable perceptions and vaping behaviours. Considering the primary goal of this research, a deliberate emphasis was placed on studies assessing e-cigarette advertisements and their engagement metrics, represented by users' comments, shares, reposts, retweets, or likes. Since users' engagement could be translated as endorsements of the promoted content, these metrics are integral components of this review. This selection criteria aimed to cover how effectively e-cigarette adverts resonated with the social media audience, ultimately shaping users' perceptions and potentially influencing vaping behaviours.

The inclusion and exclusion criteria (Table 2 below) were guided through precise definitions of the population, exposure, comparator, and outcome of interest [46]. To identify all potential studies related to the research question, the title and abstracts were initially screened independently by two reviewers (LS and GM) with any disagreements resolved with a third

**Table 2. Selection criteria for relevant studies.**

| Inclusion criteria: |
| --- |
| • All gender individuals over 10 years old. |
| • Core Anglophone countries, including the UK, Ireland, USA, Canada, Australia, and New Zealand. |
| • E-cigarette commercial promotion on public and popular social media, including marketing campaigns, advertisements and paid publicities. |
| • Studies assessing the association between the commercial promotion of e-cigarettes on social media on youths' positive perceptions about vaping and vaping behaviours. |
| • Content/thematic analysis of e-cigarette adverts on social media, assessing users' positive engagement (e.g., comments, shares, reposts, retweets, and likes). |

| Exclusion criteria: |
| --- |
| • Paediatric population and individuals from non-Anglophone countries. |
| • Tobacco-related products such as conventional cigarettes. |
| • Traditional social media platforms such as television and radio. |
| • Non-commercial promotion of electronic cigarettes on social media (e.g., non-sponsored posts and user-generated content). |
| • Studies assessing negative perceptions of vaping and the decrease in vaping behaviours. |
| • Content or thematic analysis studies of user-generated content. |
| • Content or thematic analysis studies of commercial posts with no relevant implication on positive perceptions of vaping and vaping behaviours. |

reviewer (SG). The full-text papers were again independently assessed by two reviewers with input from a third reviewer for disagreements.

## Charting the data

Aiming to create a descriptive summary for all the studies selected for inclusion, a customized data charting table was formulated according to the research questions. An iterative, collaborative, and process-oriented data process was followed during this stage [37]. A descriptive summary of the extracted studies included general bibliographic information, population description, social media assessment, study design, conflicts of interest and funding of included studies. In addition to the first data chart, a second descriptive summary of the extracted studies included the study's general bibliographic information, objectives, method, outcomes, and conclusion.

## Summarizing, synthesizing, and reporting the results

Informed by the framework developed by Levac et al (2010) [38] the last phase included data analysis, reporting the results, and applying meaning to the results. The data analysis phase comprised a descriptive numerical summary and thematic analysis of the included studies, which respectively displayed relevant information and characteristics of the included papers and contributed to the identification of patterns within the extracted data by using defining themes according to the research questions [47].

Results were reported via a narrative synthesis of the evidence, including assessing the descriptive summary, defined themes, and considerations of included studies. This process aimed to correlate the commercial promotion of electronic cigarettes on social media with vaping initiation, positive perceptions of vaping, and secondary outcomes of interest. Consideration of the findings from a broader perspective and their respective implications for prospective research was developed to advance the legitimacy of the scoping review [38].

## Results

### Descriptive summary of included studies

1385 papers were retrieved from the database searches (Fig 1). Once 614 duplicate papers were removed the title and abstracts of 771 papers were screened by three independent reviewers. 646 studies were excluded because they did not meet the inclusion criteria. Through the process of reviewing 125 full papers, a further 114 studies were excluded. In total, eleven papers met the inclusion criteria [48–58] (Table 3) as shown in the flow diagram [59] (Fig 1).

### Thematic analysis

The summary of each study's main theme, including its objective, method, main results, and key findings can be found in Table 4 below:

### Thematic analysis

**Characteristics of the products marketed on social media.** According to studies by Chu et al. (2015) [49] and Laestadius et al. (2019) [54], promoting electronic cigarette liquid, including its flavours and nicotine levels, is a marketing strategy that effectively attracts social media users. Chu et al. (2015) [49] found that Twitter users retweeted posts containing flavour-related messages more frequently than non-flavour posts, indicating that flavours are an attractive feature of e-cig promotion on social media. The authors concluded that the promotion of flavours should be monitored closely by public health authorities. Laestadius et al. (2019) [54] evaluated most in-depth young adults' perceptions of e-liquid marketing content on Instagram and found that posts promoting e-liquids were appealing to individuals across diverse smoking statuses, despite some discrepancies. For all participants in this study, the appeal of e-liquid marketing on Instagram was primarily influenced by the flavour's endorsement, the trustworthiness of the social media account, the visual design of the posts, and nicotine levels [54]. In contrast, the appeal among all smoking status groups was reduced in the presence of vaping culture language and insensibility to nicotine addiction [54]. The use of Federal Food and Drug Administration (FDA) warning statements reduced the appeal of posts among several participants, although this effect was most significant for non-tobacco users. In comparison with other smoking status groups, non-tobacco users were more attracted to posts advertising nicotine-free e-liquids with recognizable flavours. The authors conclude the findings provide valuable insights for policymakers to develop interventions that aim to reduce the appeal of e-liquid promotion, especially to non-tobacco users [54].

**Characteristics of posts.** Laestadius and colleagues (2020) [55] used the same participant sample and research methodology (focused groups) as Laestadius et al. (2019) [54] to evaluate young social media users' perceptions of e-cigarette hashtags on Instagram and their role in health communication. Results showed that for the majority of the participants, hashtags were acknowledged as containing health-related claims (e.g., #vapingsavedmylife), despite being frequently considered ambiguous, exaggerated or unrealistic. Vape and dual user participants disclosed more support for e-cigarette hashtag claims in comparison with smokers and non-tobacco users, possibly because of their experiences and positive perceptions of vaping [55]. The research findings suggest that hashtags used to promote electronic cigarette brands on Instagram often contain health-related claims and should be further explored and considered for marketing regulatory purposes [55].

The hashtag #ejuice, frequently used in the promotion of electronic cigarettes on Instagram [60] was assessed in the Dormanesh, Kirkpatrick and Allem (2020) study [51], which aimed to explore the prevalence of cartoon images promoting electronic cigarettes products and its

**Fig 1. PRISMA flowchart diagram of searched studies.**

**Table 3. Descriptive summary of study design and funding of extracted studies.**

| AUTHORS AND YEAR | STUDY DESIGN | POPULATION | SOCIAL MEDIA | CONFLICTS OF INTEREST | FUNDING |
|---|---|---|---|---|---|
| *Chu et al., 2015* | Exploratory network analysis | Not indicated, but suggestive from the U.S. | Twitter | The authors have declared that no competing interests exist | National Cancer Institute, FDA Center for Tobacco Products |
| *Chu, Sidhu and Valente, 2015* | Exploratory network analysis | Not indicated, but suggestive from the U.S. | Facebook, Twitter, Instagram and Google + | The authors have declared that no competing interests exist | National Cancer Institute, FDA Center for Tobacco Products |
| *Daniel, Jackson and Westerman, 2018* | Exploratory network analysis | Not indicated, but suggestive from the U.S. | YouTube | Not declared | Not declared |
| *Dormanesh, Kirkpatrick and Allem, 2020* | Exploratory network analysis | Not indicated, but suggestive from the U.S. | Instagram | The authors have declared that no competing interests exist | The University of California, Research Grants Program Office, Tobacco-Related Diseases Research Program |
| *Han et al., 2022* | Experimental | Youths ($N = 41$) aged 18 to 23 years old, from the U.S. | YouTube | The authors have declared that no competing interests exist | Indiana University Graduate Professional Student Government |
| *Kong et al., 2021* | Exploratory network analysis | Not indicated, but suggestive from the U.S. | Facebook | The authors have declared that no competing interests exist | National Institute on Drug Abuse |
| *Laestadius et al., 2019* | Focus Groups | Youths ($N = 69$) aged 18 to 24 years old from Milwaukee, U.S. | Instagram | The authors have declared that no competing interests exist | National Cancer Institute, FDA Center for Tobacco Products |
| *Laestadius et al., 2020* | Focus Groups | Youths ($N = 69$) aged 18 to 24 years old from Milwaukee, U.S. | Instagram | The authors have declared that no competing interests exist | National Cancer Institute FDA Center for Tobacco Products |
| *Phua, Jin and Hahm, 2018* | Experimental | Youths ($N = 141$) with a mean age of 20.5 years old, from the U.S. | Instagram | The authors have declared that no competing interests exist | The authors declared no financial support for the research or publication. |
| *Phua, Lin and Lim, 2018* | Experimental | Youths ($N = 600$) aged 18 and 34 years old. The setting is not indicated but is suggestive of the U.S. | Instagram | The authors have declared that no competing interests exist | Grady College of Journalism and Mass Communication—University of Georgia |
| *Pokhrel et al., 2021* | Cross-sectional survey | Youths ($N = 2,327$) between 18 to 25 years old, from Hawaii, U.S. | Facebook, Instagram, Twitter, YouTube, and Snapchat | The authors have declared that no competing interests exist | National Cancer Institute |

engagement. The analysis indicated that posts that included cartoons generated a higher level of user engagement (measured by likes) in comparison to posts without cartoons. The results of the study indicate that e-cigarette companies frequently utilize cartoons as a marketing strategy and that these cartoons are effective in attracting Instagram users [51].

Arguments, in the context of social media, are persuasive elements used to convince people to purchase a product and can be presented in the form of messages. Considering that Han and collaborators' (2022) [52] explored non-smoking young adults' responses to e-cigarette promotion on YouTube, by assessing participants' reactions to argument-quantity commercials (determined accordingly to the number of persuasive words present). The study revealed that a reduced number of arguments had the most significant influence on various factors, such as participants' attention, emotional arousal, user engagement (measured by the liking of advertisements), and vaping urges [52]. Authors believe that because more arguments usually contain more negative words (e.g., highlighting the risks of smoking versus vaping), it may activate non-smoking defensive message processing. Through these findings, meaningful insights can be used in favour of regulatory policies related to the promotion of e-cigarettes [52].

**Table 4. Overview of the objectives, methods, main results and key findings of extracted studies.**

| Authors and year | Objectives | Method | Main results | Key findings |
|---|---|---|---|---|
| *Chu et al., 2015* | Examine the message content of leading electronic cigarette brands (Blu and V2) on Twitter and compare the likelihood of flavour and non-flavour messaging being shared with other viewers. | Content analysis of 1180 tweets containing commercial promotion of electronic cigarettes posted by the brands Blu and V2. | Flavour-related industry content was shared significantly more often by Twitter users than non-flavour industry content. | Flavour promotion is an attractive marketing strategy used by the e-cigarette industry on social media and should be further supervised by public health authorities. |
| *Chu, Sidhu and Valente, 2015* | Evaluate marketing strategies of leading e-cigarette brands (Blue and V2) on Facebook, Twitter, Instagram and Google+, aiming to identify the affordance of each social networking site in their marketing. | Secondary analysis of Blu and V2 content on four distinct social media over two-and-a-half years, including weighting word's relevance and frequency and assessing users' interactions. | Overall, Blu displayed significantly more user interactions on the four selected social media sites in comparison with V2. While V2 focus on directing users to its website, Blu focuses on engaging users through conversations, benefiting the most from Twitter for this purpose. | E-cigarette brands benefit from social media affordance by using different marketing strategies to attract and engage a range of potential consumers. |
| *Daniel, Jackson and Westerman, 2018* | Explore the dynamics of the vaping community on YouTube, by assessing users' parasocial interactions with social media influencers. | 34 influencers' profile videos from the YouTube channel Vape Capitol were selected, and their respective viewer's comments were analysed through the SSSW components. | Sensory and social segments were the most frequent among all the SSSW. Positive parasocial interaction with social media influencers and parasocial satisfaction were identified among users' comments. | Electronic cigarette brands are benefiting from parasocial interactions between social media users and influencers to promote their products, creating a loyal audience and generating a sense of group identity. |
| *Dormanesh, Kirkpatrick and Allem, 2020* | Evaluate whether electronic cigarette brands are using cartoon-based strategies to promote their products on Instagram. | Content analysis of 1936 e-cigarette brands' promotional posts on Instagram containing a vaping-related hashtag. | 100 distinct electronic cigarette brands used cartoons on promotional posts on Instagram and displayed higher user engagement in comparison with non-cartoon ones. | The vape industry uses cartoons in its marketing strategies, appealing mostly to vulnerable consumers. |
| *Han et al., 2022* | Explore non-smokers' responses to electronic cigarette commercials on YouTube, by assessing participants' cognitive, emotional, and attitudinal reactions to argument quantity and endorsement-type content. | 41 undergraduate students' reactions and emotions to eight e-cigarette brand commercials from YouTube were analysed using a video coding system, electrocardiography and electrodermal activity. | Low-argument commercials resulted in participants' higher attention, emotional arousal, ad liking, and vaping urge. In comparison to average-citizens endorses, celebrities achieved greater participants' attention and ad liking, even when an increased number of arguments were present. However, celebrities resulted in lower emotional arousal and vape urge among participants. | Low-argument quantity commercials might have the greatest impact on vaping urges among non-smokers. |
| *Kong et al., 2021* | Identify marketing strategies of electronic cigarette brand-sponsored Facebook profiles and evaluate social media users' engagement. | Content analysis of 225 posts of 26 e-cigarette brand-sponsored profile pages on Facebook. | Photos and links were the most common types of posts, mainly featuring e-cigarette products, sales promotions, and non-sales promotional content. Posts featuring giveaways and whole devices had the highest levels of positive engagement from users. Over 40% of the posts could be accessed by underage users. | The vape industry benefits from Facebook to self-promoter by using a variety of marketing strategies, and frequently, this promotional content can be accessed by underaged individuals. |

*(Continued)*

**Table 4.** (Continued)

| Authors and year | Objectives | Method | Main results | Key findings |
|---|---|---|---|---|
| *Laestadius et al., 2019* | Evaluate young social media users' perceptions of e-liquid marketing content on Instagram. | 69 young adults aged 18 to 24 years old across diverse smoking statuses were selected to participate in a series of focus groups aiming to evaluate their perceptions of e-liquid marketing content on Instagram. | E-liquids were appealing to individuals across diverse smoking statuses, with higher intentions to use e-liquids observed among nicotine users. Nicotine-free e-liquids with recognizable flavours and appealing visual designs were more appealing to non-tobacco users, while those with FDA warning statements were discouraging. | Diverse elements such trustworthiness of the social media account, visual design, e-liquid flavours, and nicotine levels influenced were relevant when assessing the appeal assessing youths' appeal for e-liquid marketing on Instagram. |
| *Laestadius et al., 2020* | Explore how young adults interpret electronic cigarette hashtag claims on Instagram. | 69 young adults aged 18 to 24 years old across diverse smoking statuses were selected to participate in a series of focus groups aiming to identify e-cigarette hashtag-based claims and evaluate their validity and credibility. | Hashtags were recognized as health-related claims but were frequently interpreted as ambiguous and exaggerated. Increased support for hashtag claims was observed among vapers and dual users in comparison with smokers and non-tobacco users. | Hashtags used to promote electronic cigarette brands on Instagram usually contain health-related claims and should be further explored and considered for regulatory restriction. |
| *Phua, Jin and Hahm, 2018* | Assess the impact of celebrities, non-celebrities, and product-only endorsements of electronic cigarette advertisements on Instagram, on vaping attitudes and smoking intentions. | 141 students participated in a stimulus experiment by viewing three types of manipulated electronic cigarette adverts on Instagram and subsequently answered an online questionnaire. | Positive attitudes towards e-cigarettes and smoking intentions were found in celebrity-endorsed adverts. Competence, goodwill, trustworthiness, and attractiveness moderators were higher in celebrity-endorsed posts compared to non-celebrity ones. | Electronic cigarette promotion on Instagram endorsed by celebrities might increase users' attitudes and intentions toward vaping and smoking. |
| *Phua, Lin and Lim, 2018* | Explore the congruence between celebrities and vape products, consumers' risk-oriented, consumers' engagement towards the celebrity, consumers' attitudes towards the commercial, parasocial interactions and users' intention to vape on Instagram. | 600 participants between 18 and 34 years old, with diverse smoking statuses participated in a stimulus experiment on Instagram, featuring 20 e-cigarette celebrities' endorsed posts. | A high degree of congruence between celebrities and products increased positive attitudes towards the advertisement, word-of-mouth intent, and likelihood of use of e-cigarettes. Parasocial identification between participants and endorsers also affected the intention to use e-cigarettes. Current smokers and e-cigarette users showed more favourable attitudes towards celebrity-endorsed adverts. | E-cigarette brands endorsed by celebrities with a high perceived image congruency have higher chances to engage with target audiences, leading to more positive attitudes towards the ad, word-of-mouth intent, and e-cigarette use. |
| *Pokhrel et al., 2021* | Test the hypothesis that electronic cigarette content exposure on popular social media sites is associated with vaping initiation and progression among young adults. | 2327 young students from community colleges in Hawaii were recruited to participate in a survey and were followed up at a three-time-point. | High e-cigarette content exposure on social media was associated with vaping onset among naïve users and vaping escalation among those with previous vaping experience. | Exposure to e-cigarette content on social media may influence young adults' vaping onset or progression, by increasing users' positive beliefs. |

**Celebrities and social media influencers.** Celebrities were found to be a popular marketing strategy to promote e-cigarettes on social media and to positively influence young social media users' attitudes and intentions towards vaping. Phua, Jin, and Hahm (2018) [56] found that celebrity endorsement posts on Instagram resulted in higher positive attitudes towards e-cigarettes and smoking intentions compared to those endorsed by non-celebrities. Additionally, competence, goodwill, trustworthiness, and attractiveness moderators were higher in celebrity-endorsed posts [56]. Similarly, Phua, Lin, and Lim's study (2018) [57] found that high degrees of congruence (or similarity) between celebrities and products increased positive

attitudes towards the advertisement, word-of-mouth intent, and likelihood to use e-cigarettes. Findings also demonstrated that parasocial identification between participants and endorsers affected the intention to use e-cigarettes and that current smokers and e-cigarette users showed more favourable attitudes towards celebrity-endorsed adverts [57].

Han and collaborators (2022) [52] also assessed non-smokers' responses to endorsement-type messages (including celebrities and non-celebrities endorses) on YouTube. As authors hypothesized, celebrity-endorsed commercials resulted in higher levels of participants' attention in comparison to average-citizens ones. Furthermore, while an increased quantity of arguments in average-citizens-endorsed commercials resulted in a decrease in participants' ad liking, the presence of celebrities' endorsements did not have an impact on ad liking [52]. Surprisingly, the presence of celebrities in commercials resulted in lower emotional arousal and vape urge among participants, perhaps because non-smoker users are not primarily interested in e-cigarette topics [52].

Parasocial interactions also described as a consumer one-way interaction with a media personality, were explored in Daniel, Jackson and Westermann's study (2018) [50]. The authors evaluated how marketing strategies benefit from users' parasocial with social media influencers on YouTube under Taylor's Six-segment Strategy Wheel (SSSW). Daniel, Jackson and Westermann (2018) [50] used the communication framework SSSW to evaluate the content presented on YouTube messages and how it resonates with the vaping community. The results revealed that the sensory and social segments were the most frequent components among the SSSW and positive parasocial interaction and parasocial satisfaction were identified in viewers' comments on YouTube. The study findings suggest that the vape industry is benefiting from parasocial interactions between users and social media influencers, by creating a loyal community and generating a sense of group identity [50].

**Users' engagement with social media posts.**   Chu, Sidhu and Valente (2015) [48] and Kong et al. (2021) [53] investigated the marketing strategies used by popular e-cigarette brands on social media and their respective engagement with users. Despite using different approaches, both studies found that e-cigarette companies use unique marketing tactics to target and engage potential customers on social media platforms. Chu, Sidhu and Valente (2015) [48] provided an overview of the marketing strategies used by Blu and V2 across multiple social media platforms and discovered that the term "e-cig" was frequently used on the four analysed social media platforms, with the most common categories being political information, user interactions, and links. They also found that Blu had more user engagement than V2 on all platforms, suggesting that their interactive and community-oriented approach was more effective in engaging social media users [48].

Kong et al. (2021) [53] evaluated marketing strategies of e-cigarette brand-sponsored profile pages on Facebook and their respective post engagement and found that photos and links were the most common types of posts, mainly featuring e-cigarette products, sales promotions, and non-sales promotional content. The study identified that posts featuring giveaways and whole devices had the highest levels of positive engagement from users [53]. Most importantly, over 40% of the posts could be accessed by underage Facebook users. These findings suggest that diverse marketing strategies are used to promote e-cigarettes on Facebook, and restrictive measures related to tobacco control are urgently needed as vaping content can be accessed by vulnerable users [53].

**Electronic cigarette intentions, onset and escalation.**   The previous studies mentioned above have shown that the promotion of electronic cigarette on social networking sites, mediated by the presence of arguments and celebrities, were associated with vaping intentions among young adults [52, 56]. In the study conducted by Pokhrel et al. (2021) [58], the researchers also examined the impact of exposure to e-cigarette promotion on social media on

the initiation and escalation of vaping among youth. Additionally, the study investigated the role of positive e-cigarette use outcome expectancies (positive beliefs about vaping) among youths as a mediator in this relationship. Results indicated that among all social media included in the study (Facebook, Instagram, Twitter, YouTube, and Snapchat), the exposure to electronic cigarette promotion was highest on Facebook and Instagram [58]. Overall, high e-cigarette content exposure on social media was associated with vaping onset among never-e-cigarette users and vaping escalation among current vapers. The study revealed that affect regulation expectancies, which refer to beliefs that vaping can lead to positive health outcomes such as stress reduction, acted as mediators for both never-e-cigarette users and current vape participants [58]. Therefore, it is possible to conclude that electronic cigarette content on social media may influence youth vaping onset and progression, and users' expectancy beliefs should be further explored to prevent the increase in vaping behaviours [58].

## Discussion

This scoping review aimed to provide a comprehensive overview of the topic of the commercial promotion of e-cigarettes on social media, and its influence on positive perceptions of vaping and vaping in Anglophone countries. Eleven studies met the inclusion criteria for this review, comprising diverse research designs, such as focus groups, content analysis, cross-sectional studies, and experiments. The selected studies for this review primarily focused on youths, which is not surprising given that they represent the largest demographic of social media users [61]. In addition, young individuals are particularly susceptible to being influenced by exposure to e-cigarettes online, and to developing vaping behaviours [62–64].

Key features of commercial promotion of e-cigarettes on social media comprised the endorsement of celebrities, the attractiveness of e-liquids (including flavours and nicotine levels), the incorporation of cartoons, hashtag usage, trustworthiness, and user engagement. Importantly, the trustworthiness of e-cigarette brands and endorsements were significant mediators of the appeal of promotional posts related to vaping advertisements on social networking sites [54, 56].

Overall, the included studies demonstrate that e-cigarette brands combine appealing marketing strategies to attract customers on social media, significantly fostering positive perceptions of vaping and shaping vaping behaviours, particularly among young individuals. These results align with previous research that has established a correlation between exposure to electronic cigarette promotion on social media and users' attitudes towards vaping [20–23] and with other forms of e-cigarette exposure, such as television, radio, and magazines [65–69]. Furthermore, the use of celebrities promoting e-cigarettes online is likely to be reminiscent of the tobacco industry marketing practices [19].

Limitations of the studies include the U.S.-centric perspective in this research; all 11 studies explicitly state they were conducted in the U.S. or indicated this was the case. A significant number of U.S. studies dominated the screening of titles and abstracts, which could be attributed to the availability of substantial research funding at the national level for tobacco control, and because of the EVALI outbreak in 2019 [12–14], which fomented national-level research into the vaping impact on health and influenced debates on the role of e-cigarettes in tobacco control.

In the context of the U.S. EVALI epidemic outbreak in 2019 [12–14], a former (FDA) Commissioner stated that e-cigarettes pose an "existential threat" to youth. Because the FDA and the National Cancer Institute were key funders of the included studies, there may be a risk of bias towards US-based (and funded) studies finding unfavourable outcomes in the promotion of electronic cigarettes on social media [14]. Hence, the omission of a declaration of interest

and funding information observed in Daniel, Jackson and Westerman's (2018) study [50] is problematic. Conflicts of interest and funding should always be declared, as it is primordial information and an ethical principle in health research [70].

The sample size of the included studies is relatively small and non-diversified. For example, Laestadius et al. (2019) [54] and Laestadius et al. (2020) [55] shared the same small sample size of 69 participants; Han and collaborators (2022) study [52] primarily consisted of female participants who were White Caucasian; and Phua, Lin and Lim study (2018) [57] was majority consisted of Caucasian participants. The geographic disproportionality of electronic cigarette research contributes to increased discrepancies in regulatory policy development globally and highlights health inequalities. Although the included studies examined popular social media platforms like Facebook, Twitter, Snapchat, Instagram, and YouTube, they did not include TikTok, an emerging social media platform popular among youths [71], drawing attention to the need for updated and contextualized research on this topic.

An ecological validity issue was found in two experimental studies included in this scoping review [56, 57]. Respectively, the first study [56] consisted of screenshots of manipulated electronic cigarette pages on Instagram, displaying multiple pictures on each page, instead of exhibiting a regular social media feed, where participants could visualise individual photos as they scroll the screen down. Correspondingly, the second study [57] did not strive to use real adverts from the e-cigarette brand assessed for the study. Consequently, the methods used in these studies reduced the realistic scope of the experiment, by distancing participants from conventional social media performances. Lastly, in the context of assessing the influence of e-cigarette exposure on social networking sites on vaping onset and escalation among youths, only one study [21] was able to demonstrate this association, by monitoring participants over a long period.

## Recommendations for future research

Our findings suggest additional research funders across and outside the Anglosphere should prioritize this area of study, contributing to equity in health research and establishing comparative parameters among similar and different populations. Further, by assessing larger and diversified sample sizes, future research could achieve more comprehensive results among target populations. In terms of transparency, prospective studies should always be clear about the research setting, and demographic characteristics of the studied population, and state any possible conflicts of interest. Additionally, future research should prioritize cross-platform studies, ensuring a more comprehensive understating of the commercial promotion of e-cigarettes on multiple networking sites, and acknowledging the nuances specific to each site.

To enhance ecological validity, prospective experimental studies should aim to recreate realistic environments for participants and evaluate the applicability of research findings to real-life settings [72]. Therefore, future research should preferably cover original social media users' profiles, existent e-cigarette promotional content, and up-to-date versions of social networking sites. By avoiding using manipulated media (such as photoshopped images), authors can strive for greater findings' legitimacy by increasing the realistic scope of the experiment. Moreover, experimental study outcomes could be improved by conducting long-term evaluations of participants' vaping status after e-cigarette exposure on social media, consequently amplifying the existing knowledge related to e-cigarette initiation and escalation among the studied population.

## Policy and practice implications

Our findings have implications for stricter marketing regulations for e-cigarette adverts on social networking sites. Specifically, the current study indicates that policymakers should

consider establishing guidelines for e-cigarettes adverts on social media sponsored by celebrities, also enforcing disclosures on commercial posts [57]. Regulating tobacco-related content through counter-marketing messages across social networking sites may also contribute to reducing its appeal among youths and contesting misinformation [49]. Since e-cigarette promotional content is accessible to underage individuals, additional policy implications should focus on implementing and enforcing social media age restrictions [53, 57]. The vape industry frequently blurs the boundaries between commercial and public-generated content [54] which relates to other health-harming industry commercial activities on social media [73], as observed in the alcohol industry [74]. This concern is also reflected in the FDA's limited control over non-commercial profiles on social media [30]. Consequently, it is crucial to mention the demand for additional regulation of user-generated content on social media, driven by tobacco control considerations. Considering that unambiguous definitions of tobacco-related products are urgently needed [75], classifying e-cigarettes as a unique product as undertaken by Canada, may contribute to improved jurisdictions, and harm reduction [76]. Overall, it is imperative to conclude that comprehensive statutory frameworks concerning e-cigarette advertisements are urgently needed across the globe, considering the alarming danger of vaping.

## Limitations

This scoping review has several limitations. Firstly, grey literature was excluded from the search; hence, we might have missed relevant evidence related to the research topic. We also dismiss studies assessing e-cigarettes with other types of exposure, such as tobacco products, and traditional media. The application of these exclusion criteria contributed to a significant reduction in the availability of studies that could be included in this scoping review. Furthermore, we excluded user-generated content and thematic analysis of such content. We are aware that this exclusion could constrain the research scope and hinder the identification of pertinent studies, it was, however, a decision based on the primary objectives for this scoping review. Three studies included in this review were conducted more than five years ago [48–50], and their respective findings need to be considered according to the snapshot of the availability and performance of existing social media and marketing regulations at the time.

## Conclusion

Evidence from 11 U.S.-based studies suggests electronic cigarette exposure on social networking sites is linked with positive attitudes towards vaping and vaping behaviours. This is expressed by user engagement with the promotional content, parasocial interactions between social media influencers and users, intentions to use the products, vaping initiation and vaping escalation. Limitations of the evidence include narrow settings, small and homogeneous sample sizes, research transparency concerns, limited inclusion of multiple social networking sites, ecological validity concerns in experimental studies, and few longitudinal methodologies to cover vaping initiation and escalation. Future research should focus on incorporating larger and heterogeneous sample sizes, broader the research settings, assessing multiple social media, increasing the ecological validity of experimental studies, and prioritising longitudinal studies. Overall, all studies support the urgent need for stricter e-cigarette marketing regulations of electronic cigarette promotion on social media, including establishing guidelines for e-cigarette adverts sponsored by celebrities, requiring e-cigarette advertisement disclosure, implementing counter-marketing messages, and enforcing age restrictions.

## Supporting information

**S1 Appendix. PRISMA checklist.**
(DOCX)

**S2 Appendix. Search strategies.**
(DOCX)

## Author Contributions

**Conceptualization:** L. Chacon, G. Mitchell, S. Golder.

**Data curation:** L. Chacon, S. Golder.

**Formal analysis:** L. Chacon.

**Investigation:** L. Chacon, G. Mitchell.

**Methodology:** S. Golder.

**Supervision:** G. Mitchell, S. Golder.

**Writing – original draft:** L. Chacon.

**Writing – review & editing:** L. Chacon, G. Mitchell, S. Golder.

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
