## [Decision Letter · Decision Letter 0]

7 Aug 2023

PGPH-D-23-01086

The commercial promotion of electronic cigarettes on social media and its influence on positive perceptions of vaping and vaping behaviours in Anglophone countries: a scoping review

Dear Dr. Golder,

Thank you for submitting your manuscript to PLOS Global Public Health. After careful consideration, we feel that it has merit but does not fully meet PLOS Global Public Health’s publication criteria as it currently stands. Therefore, we invite you to submit a revised version of the manuscript that addresses the points raised during the review process.

We look forward to receiving your revised manuscript.

Kind regards,

Veena Sriram

Academic Editor

Journal Requirements:

1. We have noticed that you have uploaded Supporting Information files, but you have not included a list of legends. Please add a full list of legends for your Supporting Information files after the references list. 

Additional Editor Comments (if provided):

Reviewers' comments:

Reviewer's Responses to Questions

**Comments to the Author**

1. Does this manuscript meet PLOS Global Public Health’s publication criteria? Is the manuscript technically sound, and do the data support the conclusions? The manuscript must describe methodologically and ethically rigorous research with conclusions that are appropriately drawn based on the data presented.

Reviewer #1: Yes

Reviewer #2: Yes

2. Has the statistical analysis been performed appropriately and rigorously?

Reviewer #1: N/A

Reviewer #2: Yes

3. Have the authors made all data underlying the findings in their manuscript fully available (please refer to the Data Availability Statement at the start of the manuscript PDF file)?

Reviewer #1: Yes

Reviewer #2: Yes

4. Is the manuscript presented in an intelligible fashion and written in standard English?

Reviewer #1: Yes

Reviewer #2: Yes

5. Review Comments to the Author

Reviewer #1: The research question is relevant considering the proliferation of E-cigarette advertisements and the distinct set of challenges associated with regulating them. With the introduction of these products into markets, countries have struggled to regulate E-cigarettes – some have adapted or extended the existing framework to deal with these products; some have conceived new legislation in an attempt to address the distinctness of regulating E-cigarettes; others have banned them. But, overall, these policy responses have largely been shaped by the (limited) knowledge about E-cigarettes and their effects on population health.

The authors may consider (briefly) introducing the regulatory framework of E- Cigarettes, specifically with regard to advertising on social media platforms (either in the introduction or in a table), as this provides context for both, the thematic analysis as well as policy discussion that follow.

To improve clarity, the authors may consider defining ‘social media’ at the onset even prior to the “inclusion and exclusion” criteria (pg no:7,8). It may also be interesting for readers to understand the basis for selecting ‘core’ Anglophone countries for the SR. Are the authors aware of the status of research studies in other geographies as well ?

The thematic analysis culminates in a host of interesting issues that demand further analysis. With respect to the ‘policy and practice implications’ (pg no:26), authors may explain how their findings support stricter regulations? This section would benefit from illustrations and a bit more analysis. For example, it would be good to understand or even get a better sense of what it means to have strict regulations (mentioned in the conclusion as well). The 11 articles reviewed have a geographical focus, does the US have ‘strict(er)’ marketing regulations, or have other countries been able to do a better job regulating this issue ? Similarly, is it possible to address influencers/celebrities sponsorship via regulation (think about jurisdiction and other challenges associated with regulating and monitoring social media) or do we need to think of other innovative, non-regulatory approaches.

Reviewer #2: Thank you for the opportunity to review this manuscript. It is timely and its subject matter important. The article is well-written and conclusions supported. However, I have concerns over the review methodology. Particularly, the inclusion/exclusion criteria established. I believe the strategy may miss relevant articles because of the decision to exclude user-generated content/unclear application of how content/thematic are decided for inclusion. I have provided more notes below for the authors considerations.

Introduction:

- It may be worthwhile to mention the conflict of interest of some actors involved in e-cigarette policy debates. Particularly, how tobacco companies have created e-cigarette products, or invested in them (for example, JUUL).

- Citation error, third paragraph, first sentence.

- Context on regulatory requirements of vaping advertisers, or social media requirements of advertisers, would helpfully contextualize how these activities occur.

Methods:

- A rationale is needed to explain why non-commercial promotion of electronic cigarettes was excluded (non-sponsored posts, user generated content). Often times advertisements are not declared on social media. Regulatory agencies have warned many companies for not declaring ads. Notably, vaping products are found to not declare their ads as ads, or be compliant with standards (for example, https://www.ncbi.nlm.nih.gov/pmc/articles/PMC9869128/). Other studies, such as those assessing user-generated content, find at times commercial origins or undertones (for example, https://gh.bmj.com/content/7/6/e009112). This search strategy seemingly would not capture such activities.

- Similarly, why were some content or thematic analysis studies excluded? This type of research is a major contributor to understand how health-harming products are portrayed online to social media users. It is unclear the difference between the content/thematic studies excluded, versus those included. A rationale/explanation would be helpful.

Discussion:

- Reference error, page 23 first paragraph, last sentence.

- May be worthwhile to explore how vaping compares to other health-harming industry commercial activities on social media. Is vaping unique? Broader issues of social media or vaping-specific regulatory absences?

6. PLOS authors have the option to publish the peer review history of their article (what does this mean?). If published, this will include your full peer review and any attached files.

**Do you want your identity to be public for this peer review?** For information about this choice, including consent withdrawal, please see our Privacy Policy.

Reviewer #1: No

Reviewer #2: No

---

## [Decision Letter · Decision Letter 1]

11 Dec 2023

The commercial promotion of electronic cigarettes on social media and its influence on positive perceptions of vaping and vaping behaviours in Anglophone countries: a scoping review

PGPH-D-23-01086R1

Dear Dr Golder,

We are pleased to inform you that your manuscript 'The commercial promotion of electronic cigarettes on social media and its influence on positive perceptions of vaping and vaping behaviours in Anglophone countries: a scoping review' has been provisionally accepted for publication in PLOS Global Public Health. Thank you for your excellent work in addressing the reviewer comments.

Best regards,

Veena Sriram

Academic Editor

Reviewer Comments (if any, and for reference):

Reviewer's Responses to Questions

**Comments to the Author**

1. If the authors have adequately addressed your comments raised in a previous round of review and you feel that this manuscript is now acceptable for publication, you may indicate that here to bypass the “Comments to the Author” section, enter your conflict of interest statement in the “Confidential to Editor” section, and submit your "Accept" recommendation.

Reviewer #1: All comments have been addressed

Reviewer #2: All comments have been addressed

2. Does this manuscript meet PLOS Global Public Health’s publication criteria? Is the manuscript technically sound, and do the data support the conclusions? The manuscript must describe methodologically and ethically rigorous research with conclusions that are appropriately drawn based on the data presented.

Reviewer #1: Yes

Reviewer #2: Yes

3. Has the statistical analysis been performed appropriately and rigorously?

Reviewer #1: N/A

Reviewer #2: N/A

4. Have the authors made all data underlying the findings in their manuscript fully available (please refer to the Data Availability Statement at the start of the manuscript PDF file)?

Reviewer #1: Yes

Reviewer #2: Yes

5. Is the manuscript presented in an intelligible fashion and written in standard English?

Reviewer #1: Yes

Reviewer #2: Yes

6. Review Comments to the Author

Reviewer #1: Thank you for addressing the comments, the manuscript looks good. I am sure it will be a tremendous contribution to the existing body of work addressing issues related to E-cigarettes.

Reviewer #2: The manuscript looks excellent and my comments from the previous round of revision were addressed. I look forward to seeing this paper published.

7. PLOS authors have the option to publish the peer review history of their article (what does this mean?). If published, this will include your full peer review and any attached files.

**Do you want your identity to be public for this peer review?** For information about this choice, including consent withdrawal, please see our Privacy Policy.

Reviewer #1: No

Reviewer #2: No
